# Clinical and bioethical implications of health care interruption during the COVID-19 pandemic: A cross-sectional study in outpatients with rheumatic diseases

**Guillermo A. Guaracha-Basáñez**[1], **Irazú Contreras-Yáñez**[1], **Gabriela Hernández-Molina**[1], **Anayanci González-Marín**[1], **Lexli D. Pacheco-Santiago**[1], **Salvador S. Valverde-Hernández**[1], **Ingris Peláez-Ballestas**[2], **Virginia Pascual-Ramos**[1]*

1 Department of Immunology and Rheumatology, Instituto Nacional de Ciencias Médicas y Nutrición Salvador-Zubirán (INCMyN-SZ), Mexico City, Mexico, 2 Rheumatology Unit, Hospital General de México "Dr. Eduardo Liceaga", Mexico City, Mexico

☯ These authors contributed equally to this work.

* virtichu@gmail.com

## Abstract

### Background

To determine the impact of health care interruption (HCI), on clinical status of the patients reincorporated to an outpatient clinic for rheumatic diseases (OCDIR), from a tertiary care level center who was temporally switched to a dedicated COVID-19 hospital, and to provide a bioethical analysis.

### Methods

From March to June 2020, the OCDIR was closed; since June, it is limited to evaluate 25% of the ongoing outpatients. This cross-sectional study surveyed 670 consecutive rheumatic outpatients between June 24th and October 31th, concomitant to the assessment of the rheumatic disease clinical status by the attendant rheumatologist, according to disease activity level, clinical deterioration and adequate/inadequate control. Multiple logistic regression analysis identified factors associated to HCI and to clinical deterioration.

### Results

Patients were middle-aged females (86.7%), with median disease duration of 10 years, comorbidity (38.5%) and 138 patients (20.6%) had discontinued treatment. Primary diagnoses were SLE and RA, in 285 (42.5%) and 223 (33.3%) patients, respectively.

There were 344 patients (51.3%) with HCI. Non-RA diagnosis (OR: 2.21, 95%CI: 1.5–3.13), comorbidity (OR: 1.7, 95%CI: 1.22–2.37), patient's need for rheumatic care during HCI (OR: 3.2, 95%CI: 2.06–4.97) and adequate control of the rheumatic disease (OR: 0.64, 95%CI: 0.45–0.9) were independently associated to HCI. There were 160 patients (23.8%) with clinical deterioration and associated factors were disease duration, substantial disease activity previous HCI, patients need for rheumatic care and treatment discontinuation.

**Data Availability Statement:** All relevant data are within the paper and its Supporting information files.

**Funding:** The authors received no specific funding for this work.

**Competing interests:** The authors have declared that no competing interest exist.

## Conclusions

HCI during COVID-19 pandemic impacted course of rheumatic diseases and need to be considered in the bioethical analysis of virus containment measures.

## Introduction

The Coronavirus Disease 2019 (COVID-19) pandemic has emerged as an unprecedented challenge to health care systems and to clinicians who have been forced to adapt to health-related decisions [1, 2]. As part of virus containment measures, partial or complete closure of outpatient clinics has been implemented in health care facilities in many countries, which has impacted the management of chronic conditions, such as rheumatic diseases [3, 4]. Meanwhile, telemedicine is being incorporated to daily medical practice, but its immediate implementation has been challenging, and its impact on the patient-doctor relationship, albeit promising, is still uncertain [5–7]. The patient-doctor relationship is highly valuable in itself, with the potential to impact patients' outcomes [8], and it is the foundation for bedside clinical ethics [9].

Rheumatologists are considered essential physicians for patients with rheumatic diseases, and early access to these specialists is considered imperative in order to achieve appropriate outcomes. In fact, differences in access to subspecialty care contribute to the known disparities in morbidity and mortality from some of the rheumatic diseases. While expediting the diagnosis and treatment of rheumatic diseases reduces disparities, it has ethical implications [9–11].

The COVID-19 pandemic has led to issues in public health ethics, where the need to serve patients with COVID-19 has translated into rationing/delaying the care of patients with chronic conditions and perpetuating disparities. Chronic non-heritable conditions are considered indicators of poor health, which is associated with low income and limited access to universal health care, poor education, and minority status [1]. There is accumulating evidence that the COVID-19 pandemic might result in additional collateral damage for patients with chronic conditions due to issues in medication supply and economic setbacks to the society [11–13]. All of these factors have led to a humanitarian crisis in Latin America, where public life is characterized by fragile health systems and long-standing and pervasive inequity [14].

The Instituto Nacional de Ciencias Médicas y Nutrición Salvador Zubirán (INCMyN-SZ) is a national referral center for rheumatic diseases located in México City, where more than 7,000 patients with different rheumatic diagnosis receive health care from 21 rheumatologists/ trainees in rheumatology. In March 2020, the Mexican government declared our Institution a dedicated COVID-19 hospital, and visits to the outpatient clinic of the Department of Immunology and Rheumatology (OCDIR) were interrupted and moved to phone medical consultations [15]. Nonetheless, given the middle-low socioeconomic status of most of our patients and limited resources available at our Institution, the move has been challenging. Since June 2020, the OCDIR has been reinstalled, although only 25% of the scheduled patients currently receive face-to-face consultations.

Rheumatologists have built a strong relationship with their patients and have a privileged position during this pandemic to educate them, share international recommendations regarding immunosuppressive drugs [16], and prevent unnecessary fears that might cause patients to withdraw immunosuppressive drugs and trigger an increase in disease activity [17]. To achieve this, convenient access to rheumatologists should be guaranteed. In addition, the shortage in

medication supply might additionally impact the disease activity status of patients with rheumatic diseases; therefore, rheumatologists can provide treatment alternatives [13, 18–20].

The primary objective of this study was to identify the proportion of patients with rheumatic diseases whose access to rheumatic health care at the OCDIR of the INCMyN-SZ was affected and to determine the impact of health care interruption (HCI) on the clinical status of the underlying rheumatic disease. The secondary objectives were to explore patients' perception of access to health care and communication with the rheumatologist, and modifications made to rheumatic disease-related treatment by the patients, with the underlying reasons, during the early months of the COVID-19 pandemic. The study was registered in clinicaltrials.gov (NCT04557358). The results will be discussed from a bioethical perspective.

## Materials and methods

### Ethics

The study was performed in compliance with the Helsinki Declaration [21]. The Research Ethics Committee of the INCMyN-SZ approved the study (reference number: IRE-3467). Written informed consent was obtained from all the patients.

### Study characteristics and target population

The study had a cross-sectional design and consisted of a survey administered through June 24th to October 31st, to all the patients with a known rheumatic disease based on the diagnosis of the attendant rheumatologist, who had a face-to-face consultation when the OCDIR was reinstalled.

### Survey development and validation, and pilot testing

**Survey development.** The survey content was proposed by a committee consisting of two rheumatologists, two general physicians, and one social worker. The committee agreed on five components to be included in the survey, and subsequently, on individual items, their scale responses, and their distribution into the five components. The first version of the survey ($SV_1$) was thus generated (Table 1).

**Survey validation.** Experts determined the face and content validity of the $SV_1$. The expert committee consisted of 11 certified rheumatologists assigned to the OCDIR, who were blinded to each other's evaluation, and scored the following characteristics on standardized formats: relevance and pertinence of individual items to the survey purpose, adequate wording, appropriate language and meaning regarding individual items and instructions, and adequacy of the item's scale response. Consequently, items 1, 7, 8, 9, 16, and 17 were modified, and additional options for item scale response were included for items 7 and 9. In all cases, at least 80% agreement among experts was deemed necessary to approve the modifications; finally, the $SV_2$ was generated with 30 items (Table 1).

**Pilot testing.** A pilot test was performed in 40 consecutive outpatients from the OCDIR, who were interviewed by two coauthors to assess their perception of instruction clarity, adequacy of wording and meaning of the items and scale responses of the $SV_2$. Standardized formats were used.

Patients agreed on instruction clarity (90%), adequacy of wording and meaning of the items (90%), and adequacy of the scale response (95%). No modifications were needed, and $SV_3$ was generated for the final application (Table 1).

**Table 1. Survey versions, components and number of items.**

| Survey components | *SV$_1$ | *SV$_2$ (Post-validation) | *SV$_3$ (Post-pilot testing) |
|---|---|---|---|
| 1.- HCI and reasons | 1 | 2 | 2 |
| 2.- Patient's need for medical care and for hospitalization | 3 | 3 | 3 |
| 3. Patient's need for communication with OCDIR rheumatologists/trainees and/or additional health-care professionals from the Institution | 6 | 6 | 6 |
| 4.- Patients modification of rheumatic disease-related treatment and reasons | 3 | 3 | 3 |
| 5.- Patients perception of risk for SARS-CoV-2 infection[1] | 16 | 16 | 16 |
| **Total N° of items** | 29 | 30 | 30 |

SV = Survey version. HCI = Health care interruption.

[1]Data would not be presented in the current paper.

*Number of items.

## Definitions

*HCI* was defined as the cancelation of a scheduled face-to-face appointment to the OCDIR, without re-scheduling (either face-to-face or telephone consultation) within the next 3 months AND/OR care not provided to patients who required rheumatologic emergency care (either face-to-face or telephone consultation), AND/OR patients' decision not to attend the OCDIR.

*Non-persistence* (NP) was defined as treatment discontinuation of the medications pre-scribed for the rheumatic disease for at least 1 week.

## Assessment of the clinical status of rheumatic disease

The survey (S1 Appendix) was administered on the same day that the patients attended the OCDIR. Attendant rheumatologists were asked to score patient's clinical status immediately after the consultation. In all the cases, standardized formats were used and included four categories intended to comprehensively address the clinical status of the underlying rheumatic disease (Table 2): *the level of disease activity*, *the course of disease activity*, which was defined

**Table 2. Categories related to the rheumatic disease clinical status and the corresponding pre-specified criteria.**

| Categories | Sub-categories | Pre-specified criteria* |
|---|---|---|
| **The level of disease activity (at the current evaluation)** | Without disease activity | No symptoms AND no clinical findings AND relevant serological markers within normal values. |
| | Substantial disease activity level | Two out of 3 of the following: Symptoms, clinical findings, relevant serological markers. |
| **The (current) course of disease activity** | Clinical deterioration | Worsening of pre-existing symptoms ± new symptoms AND/OR worsening of pre-existing clinical findings ± new clinical findings AND/OR worsening of pre-existing abnormal serological markers ± new relevant abnormal serological markers. |
| | Similar disease activity level | Similar symptoms AND similar clinical findings AND similar values of serological markers. |
| | Clinical improvement | Improvement in symptoms AND/OR in clinical findings AND/OR in relevant abnormal serological markers. |
| | Still in remission | No symptoms AND no clinical findings AND relevant serological markers within normal values at both evaluations (current and previous to HCI). |
| **The rheumatic disease control** | Adequate control of the rheumatic disease | Symptoms (if any) AND clinical findings (if any) AND serological markers, within an acceptable target AND that do not required treatment adjustment. |
| | Inadequate/Insufficient control of the rheumatic disease | Two out of 3 of the following: Symptoms, clinical findings, relevant serological markers, out of target, and that require treatment adjustment. |

*Used by the independent observer to avoid variability in the evaluation of the rheumatic disease clinical status.

considering the current level of disease activity compared to the level of disease activity at the last consultation before HCI, and *the rheumatic disease control*. The last category recorded rheumatologist treatment recommendations at the end of the consultation. The four categories and sub-categories were proposed after at least 80% of the 11 certified rheumatologists assigned to the OCDIR agreed on them.

In addition, an independent observer who was a certified rheumatologist, reviewed all the charts from the patients included, and scored the categories related to the clinical status of the underlying rheumatic disease, according to the pre-specified criteria summarized in Table 2, twice, at the last consultation before HCI and at the first face-to face consultation after the OCDIR was reinstalled. Pre-specified criteria included symptoms, clinical findings and serological markers, available in the medical charts; these were identified, analyzed and integrate by the independent observer, who additionally considered the specific diagnosis of the underlying rheumatic disease.

## Sample size calculation

The sample size for pilot testing was 40 patients, as per the recommendations for pilot testing [22]. For the survey application, we obtained a sample size of at least 346 patients. We estimated that 42% of the patients from the OCDIR would experience HCI [19].

## Statistical analysis

Descriptive statistics were used with frequencies and percentages for dichotomous variables, and mean ± standard deviation (SD) or median (Q25-Q75) were used for continuous variables with normal and non-normal distributions, respectively. The characteristics of patients with and without HCI, and with/without clinical deterioration were compared using $X^2$ test for the categorical variables, Student' s t-test for continuous variables with a normal distribution, and the Mann–Whitney U test for continuous variables with non-normal distribution. The face validity and content validity of the $SV_1$ were determined by experts, with agreement percentages of $\geq$80%.

Stepwise backward multiple logistic regression analysis was performed to identify factors associated with HCI and with clinical deterioration. Variables included in the models tested were selected according to their statistical significance in the univariate analysis ($p \leq 0.20$). Previously, correlations between specific variables were analyzed, and when the Pearson correlation coefficient was $\geq$0.75, they were selected for inclusion in the model.

In order to avoid variability due to the assessment of the clinical status of the underlying rheumatic disease by 21 rheumatologists, the independent observer scored the categories as previously described. Missing data varied from 0% to 27.3% (for the assessment of disease activity level at the last consultation before HCI). No imputation was done.

All statistical analyses were performed using SPSS (version 21.0, IBM Corp., Armonk, NY, USA) and STATA (version 14.0). A value of p <0.05 was considered statistically significant.

## Results

### Population characteristics

During the study period, 672 patients completed the $SV_3$; two patients missed component 1 that assessed the primary objective and thus were discarded. The 670 surveys corresponded to 90% of the patients with a scheduled consultation who visited the OCDIR.

Patient characteristics are summarized in Table 3. Briefly, patients were primarily middle-aged females (86.7%), with 12 (9–17) years of education. The majority of the patients had

**Table 3. Characteristics of the population and their comparison in the subpopulations defined according to HCI/non-HCI.**

| | Overall population | HCI | Non-HCI | p |
|---|---|---|---|---|
| | N = 670 | N = 344 | N = 326 | |
| **Socio-demographic characteristics** | | | | |
| Females* | 581 (86.7) | 301 (87.5) | 280 (85.8) | 0.54 |
| Age, years | 46 (35–57) | 43 (33–56) | 47.5 (36–58) | 0.06 |
| Living together* | 301 (44.9) | 149 (43.3) | 152 (46.6) | 0.39 |
| Years of scholarship | 12 (9–17) | 12 (9–17) | 12 (9–17) | 0.07 |
| Access to Social Security* | 211 (31.5) | 106 (30.8) | 105 (32.2) | 0.70 |
| **Rheumatic disease characteristics** | | | | |
| SLE diagnosis* | 285 (42.5) | 156 (45.3) | 129 (39.6) | <0.001 |
| RA diagnosis* | 223 (33.3) | 83 (24.1) | 140 (42.9) | <0.001 |
| Other rheumatic diagnosis* | 162 (24.2) | 105 (30.5) | 57 (17.5) | <0.001 |
| Disease duration, years | 10 (5–18) | 10 (4.5–18) | 11 (6–18) | 0.05 |
| N° of DMARDs/patient (±SD) | 1.6 (1) | 1.7 (1.0) | 1.6 (0.9) | 0.98 |
| Rheumatic disease comorbidity index score ≥1* | 265 (39.5) | 157 (45.6) | 108 (33.1) | 0.001 |
| **Survey components** | | | | |
| Patient's need for rheumatic care* (6 MD) | 137 (20.4) | 104 (30.4) | 33 (10.2) | <0.001 |
| NP with rheumatic disease-related treatment* (24 MD) | 138 (21.3) | 82 (24.4) | 56 (18) | 0.045 |
| **Rheumatic disease clinical status** | | | | |
| Substantial disease activity level* | 274 (40.9) | 162 (47) | 112 (34.4) | 0.001 |
| Adequate control of the rheumatic disease* | 448 (66.8) | 209 (60.7) | 239 (73.3) | 0.001 |
| Clinical deterioration* | 160 (23.8) | 91 (26.4) | 69 (21.2) | 0.109 |

*Number (%) of patients, data presented as median (Q25-Q75) unless otherwise indicated. SLE = Systemic Lupus Erythematosus. RA = Rheumatoid Arthritis. DMARDs = Disease modifying anti-rheumatic drugs. SD = Standard deviation. HCI = Health Care Interruption. MD = Missing data. NP = Non-persisting with treatment.

systemic lupus erythematosus (SLE) (42.5%) and rheumatoid arthritis (RA) (33.3%), and the remaining patients had 17 additional rheumatic diagnoses (S1 Table). Disease duration was 10 years (5–18 years), and 39.5% of the patients had a comorbidity.

## Survey results

During the first months of the pandemic, 344 patients (51.3%) experienced HCI, and the most frequent reasons are summarized in Fig 1. Meanwhile, 257 patients (38.4%, 6 missing data)

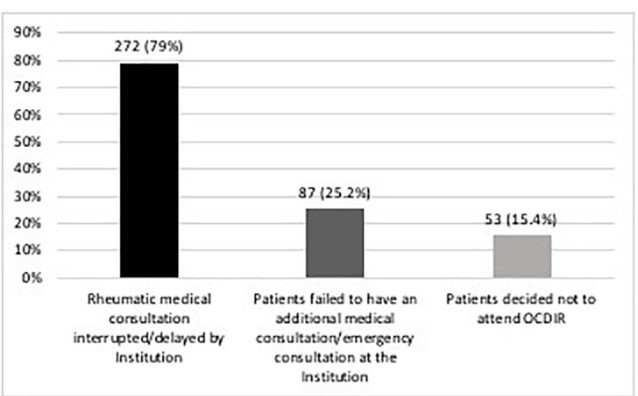

**Fig 1. Distributions of the reasons referred by 344 patients with HCI.**

required medical care, of whom 137 patients (53.3%) required medical care related to their primary rheumatologic diagnosis. In addition, 34 patients (5.1%, 5 missing data) required hospitalization, among whom 16 (47.1%) required rheumatic disease-related hospitalization.

In addition, 156 patients (23.3%, 5 missing data) and 97 patients (14.5%) needed to communicate with their primary rheumatologist because of concerns related to their rheumatic disease and concerns related to rheumatic disease associated treatment, respectively.

Finally, 404 patients (60.3%, 24 missing data) were found to be compliant with the prescribed treatment. Of the remaining 242 patients, 138 patients (57%) discontinued treatment, primarily because it was unavailable (70.3%).

## Assessment of the clinical status of rheumatic disease

At the OCDIR, all the patients were assessed: 66.8% had the rheumatic disease under adequate control, 40.9% had substantial disease activity level, and 23.8% showed clinical deterioration (compared to last clinical assessment before OCDIR closure) (Table 3).

## Impact of HCI on the clinical status of rheumatic disease

The 344 patients who experienced HCI were compared to those who did not, and the results are summarized in Table 3. Briefly, patients from the former group had more frequent non-RA diagnosis (vs. RA diagnosis), shorter rheumatic disease duration, more frequent comorbidity [23], referred more frequent need for rheumatic care, were more frequent NP with medications, had more frequent disease activity, and less frequent adequate control of the underlying rheumatic disease than patients included in the latter group.

In the regression analysis, we included the following variables: age, years of formal education, non-RA diagnosis, years of rheumatic disease duration, comorbidity, patient's need for rheumatic care (included in survey component 2), NP with rheumatic disease-related treatment (included in the survey component 4), rheumatic disease control (highly correlated to the level of disease activity), and course of disease activity. As summarized in Table 4, non-RA diagnosis, comorbidity, and patient's need for rheumatic medical care were risk factors independently associated with HCI, while adequate control of the disease was a protective factor. Regression analysis was repeated with rheumatic disease control and the course of the disease defined by the independent observer; the results were similar to those previously described (S2 Table).

We further compared patients with clinical deterioration (N = 160) to those without (N = 510). The results are summarized in Table 5. Patients from the former group had longer rheumatic disease duration, referred more frequent need for rheumatic care, and had more frequent NP with treatment than their counterparts. In addition, patients from the former group had more frequent substantial disease activity level at the last consultation before HCI, which was defined by the independent observer.

**Table 4. Regression analysis to identify factors associated with HCI.**

|  | OR | 95% CI | p |
|---|---|---|---|
| Non-RA diagnosis | 2.21 | 1.5–3.13 | ≤0.001 |
| Rheumatic disease comorbidity index score ≥1 | 1.70 | 1.22–2.37 | 0.002 |
| Patient's need for rheumatic medical care | 3.2 | 2.06–4.97 | ≤0.001 |
| Adequate control of the rheumatic disease | 0.64 | 0.45–0.9 | 0.013 |

OR = 0dds Ratio; CI = confidence interval; RA = rheumatoid arthritis. $R^2$ = 0.085

**Table 5. Comparison of characteristics between patients with clinical deterioration and their counterpart.**

| | Patients with clinical deterioration | Patients without clinical deterioration | p |
|---|---|---|---|
| | N = 160 | N = 510 | |
| Socio-demographic characteristics | | | |
| Females* | 140 (87.5) | 441 (86.7) | 0.73 |
| Age, years | 46 (36.5–58) | 46 (34–57) | 0.44 |
| Living together* | 70 (43.7) | 231 (45.2) | 0.73 |
| Years of scholarship | 12 (9–17) | 12 (9–17) | 0.53 |
| Access to Social Security | 46 (28.7) | 165 (32.3) | 0.39 |
| Rheumatic disease characteristics | | | |
| SLE diagnosis* | 66 (41.2) | 219 (42.9) | 0.26 |
| RA diagnosis* | 61 (38.1) | 162 (31.7) | 0.26 |
| Other rheumatic diagnosis* | 33 (20.6) | 129 (25.2) | 0.26 |
| Disease duration, years | 12 (6–20) | 10 (5–18) | 0.05 |
| N° of DMARDs/patient (±SD) | 1.7 (1) | 1.6 (1) | 0.86 |
| Rheumatic disease comorbidity index score ≥1* | 65 (40.6) | 200 (39.2) | 0.75 |
| Patients with substantial disease activity level previous HCI*± | 53 (33.8) | 114 (23.1) | 0.008 |
| Survey components | | | |
| HCI* | 91 (56.8) | 253 (49.6) | 0.10 |
| Patient's need for rheumatic medical care* (6 MD) | 45 (28.1) | 92 (18.2) | 0.007 |
| NP with rheumatic disease-related treatment* (24 MD) | 45 (28.1) | 92 (18.2) | 0.007 |

*Number (%) of patients, data presented as median (Q25-Q75) unless otherwise indicated. SLE = Systemic Lupus Erythematosus. RA = Rheumatoid Arthritis. DMARDs = Disease modifying anti-rheumatic drugs. SD = Standard deviation. HCI = Health Care Interruption. MD = Missing data. NP = Non-persisting with treatment.

The following variables were included in the multiple regression logistic analysis that was used to investigate factors associated to clinical deterioration, considered the dependent variable: years of rheumatic disease duration, substantial disease activity before HCI, HCI, patient's need for rheumatic care, and NP with treatment. Disease duration (OR: 1.02, 95%CI: 1.00–1.04, p = 0.011), substantial disease activity level previous HCI (OR: 1.63, 95%CI: 1.08–2.46, p = 0.019), need for rheumatic care (OR: 1.65, 95%CI: 1.07–2.55, p = 0.022), and NP with treatment (OR: 1.61, 95%CI: 1.04–2.49, p = 0.030) were associated with clinical deterioration ($R^2$ = 0.034).

## Discussion

In May 2020, the World Health Organization (WHO) conducted a survey in 155 countries and found that the COVID-19 pandemic had disrupted health services to patients with non-communicable diseases [24]. The study confirmed the findings of the WHO in a substantial number of Mexican outpatients with chronic rheumatic diseases. We found that one in two patients experienced HCI, in line with results of previous reports from the United States [19, 25] and Europe [3, 26], although higher percentages had been described among surveyed patients attending rheumatology services in 35 European countries [27], and attending a general rheumatology clinic at designated COVID-19 hybrid hospital in Malaysia [28]. The reasons for medical care interruption highlight a combination of institutional decisions to temporally close outpatient clinics and patients' own decisions to avoid hospital visits [19, 25, 27, 28].

One major clinical and ethical concern, is the association between HCI and patients' outcomes. Our study showed that adequate control of rheumatic disease was protective and associated with HCI. Few studies have addressed this topic, with conflicting results. Michaud et al. [19] conducted a COVID-19 questionnaire survey during the latter half of March 2020 in 7061 patients with rheumatic diseases; among the 530 US respondents, 42% reported some changes in their care in the previous 2 weeks, and responders with high disease activity were likely to report canceled or postponed appointments. Endstrasser et al. [29] found that the COVID-19 lockdown limited the physical activity of 63 patients with end-stage hip and knee osteoarthritis and candidates for joint replacement, and had a negative impact on patients' pain and physical function. Meanwhile, Ciurea et al. [3] demonstrated that a transient reduction in the rheumatology services during the COVID-19 pandemic had no detrimental impact on the disease course in 666 Swiss patients with axial spondyloarthritis, RA and psoriatic arthritis; however, the authors reported a 129% increase in the number of remote assessments, in an effort to compensate for the drop of the face-to face consultations, that might have influenced patients' outcomes.

The study found additional factors independently associated with HCI, namely non-RA diagnosis, comorbidity, and patient's need for rheumatic medical care. A plausible explanation might be related to patients with the above characteristics being considered candidates for tight control and to more frequent medical supervision that might translate into a higher risk of having their medical appointments interrupted during the pandemic.

(Long) disease duration, substantial disease activity level previous HCI, need for rheumatic care, and NP with (rheumatic disease-related) treatment were associated with clinical deterioration. Hassen et al. [13–30] surveyed 3000 rheumatic patients during the current pandemic in Saudi Arabia, with the aim to capture patients' experiences. Among the 637 respondents, patient perception of worsened disease activity was correlated to unplanned healthcare visits (which might be considered a surrogate for the need for rheumatic medical care), medication non-adherence (a surrogate for NP with treatment), and difficulty in accessing medication, which was in fact the main reason for treatment discontinuation among our patients. The Egyptian College of Rheumatology surveyed 1037 patients with RA to assess the impact of the first wave of the pandemic [30]. The authors reported that up to 41% of the patients referred difficulty in obtaining hydroxychloroquine and 40.7% of the patients referred disease flare. Non-adherence to treatment has been extensively described in patients with rheumatic diseases during the COVID-19 pandemic and has been attributed to limited access to medications as well as to patients' fears related to immunosuppressive drugs [3, 13, 19, 25, 26, 28, 30–32].

The ethical implications of the results from the present study should be addressed from the existing tension between public health ethics and clinical ethics [1]. As clinicians, rheumatologists are trained to serve individual patients, and patient-centered care has been proposed as the optimal conceptual model of care for patients affected by rheumatic diseases [33, 34]; it is founded on the base of patient-doctor relationship, which is characterized by trust and guided by physician concern for the patient's best interest [35]. Meanwhile, the ethical dimensions of the pandemic have pushed clinicians to consider the greater patient community's demand and hold back on individual patient care [1]. This change in medical practice is challenging our morals because of competing obligations, with a high risk of moral distress and moral injury [36, 37]. Clinical equipoise, which remains an ethical condition for clinical trials, is defined as a state of genuine uncertainty on the part of the clinical investigator regarding the comparative therapeutic merits of each arm in a trial [38]. The results of our study showed that patients in need of health care follow-up might be adversely impacted in the absence of medical appointments, and do not support the condition of clinical equipoise, so delaying their clinical care might be unacceptable.

What are we, as individual rheumatologists morally required to do, given the circumstances surrounding the pandemic? International ethical codes related to physicians' responsibilities

in disaster response, call to serve the patients in most need *and continue to serve the usual patients* [39]. The consideration of usual patients was recently highlighted by Ezequiel et al. [40], who proposed that there should be no difference in allocating scarce resources between COVID-19 patients and those with other medical conditions. We agree with Feldman et al. [41], more than ever, we now need to consider our most vulnerable rheumatology patients, particularly those at high risk of negative outcomes. As rheumatologists, we need to be accessible to them. Telemedicine and virtual consultation can improve access to specialist care [11], but all patients may not have reliable phone or internet access, especially in the Latin American region. Also, as recently highlighted by Panush et al [42], "substituting interaction by telephone call for a hand-on visit sadly misses most of the intrinsic elements of compleat patient care . . . and contravenes our traditional notions of patient care".

This study has limitations that need to be considered. This is a single-center study, and the rheumatic patient's local community might have particular characteristics. The survey application might have missed rheumatic patients with COVID-19 and/or hospitalized (for any reason) during the study period. Clinical status was assessed by 21 rheumatologists with variable levels of expertise, but similar results were obtained when a single evaluator scored the clinical status. Self-reported answers may be subject to various biases. Lastly, the study was not designed to establish causality.

## Conclusions

During the COVID-19 pandemic, HCI affected a substantial number of patients with rheumatic conditions and impacted the disease course. The negative impact might be related to the postponement of care for the most vulnerable patients, those in need for health care provision, previous substantial disease activity, and the difficulty in accessing prescribe medications. Global health initiatives to address the burden of the COVID-19 pandemic on chronic conditions need to be applied with a regional approach. Finally, as individuals, each one of us should be mindful of our moral and ethical codes that should not be violated by our actions.

## Supporting information

**S1 Appendix. COVID-19 survey.**
(PDF)

**S1 Table. N˚ (%) of patients with a face-to-face consultation at the OCDIR, with the ten most frequent rheumatic diagnoses specified.**
(PDF)

**S2 Table. Regression analysis to identify factors associated with HCI.**
(PDF)

## Acknowledgments

We acknowledge all the clinicians from the department of Immunology and Rheumatology of the Instituto Nacional de Ciencias Médicas y Nutrición Salvador Zubirán, for their support.

## Author Contributions

**Conceptualization:** Guillermo A. Guaracha-Basáñez, Irazú Contreras-Yáñez, Gabriela Hernández-Molina, Anayanci González-Marín, Lexli D. Pacheco-Santiago, Salvador S. Valverde-Hernández, Ingris Peláez-Ballestas, Virginia Pascual-Ramos.

**Data curation:** Irazú Contreras-Yáñez, Anayanci González-Marín.

**Formal analysis:** Guillermo A. Guaracha-Basáñez, Irazú Contreras-Yáñez.

**Investigation:** Guillermo A. Guaracha-Basáñez, Gabriela Hernández-Molina, Anayanci González-Marín, Lexli D. Pacheco-Santiago, Salvador S. Valverde-Hernández, Ingris Peláez-Ballestas.

**Methodology:** Irazú Contreras-Yáñez, Virginia Pascual-Ramos.

**Resources:** Virginia Pascual-Ramos.

**Software:** Irazú Contreras-Yáñez.

**Supervision:** Guillermo A. Guaracha-Basáñez, Ingris Peláez-Ballestas, Virginia Pascual-Ramos.

**Validation:** Guillermo A. Guaracha-Basáñez, Irazú Contreras-Yáñez, Gabriela Hernández-Molina, Ingris Peláez-Ballestas, Virginia Pascual-Ramos.

**Visualization:** Guillermo A. Guaracha-Basáñez, Ingris Peláez-Ballestas, Virginia Pascual-Ramos.

**Writing – original draft:** Virginia Pascual-Ramos.

**Writing – review & editing:** Guillermo A. Guaracha-Basáñez, Irazú Contreras-Yáñez, Gabriela Hernández-Molina, Anayanci González-Marín, Lexli D. Pacheco-Santiago, Salvador S. Valverde-Hernández, Ingris Peláez-Ballestas, Virginia Pascual-Ramos.

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
