## [Decision Letter · Decision Letter 0]

7 Apr 2021

PONE-D-21-03232

Clinical and bioethical implications of health care interruption during the Covid-19 pandemic: A cross-sectional study in outpatients with rheumatic diseases.

PLOS ONE

Dear Dr. Pascual-Ramos,

Thank you for submitting your manuscript to PLOS ONE. After careful consideration, we feel that it has merit but does not fully meet PLOS ONE’s publication criteria as it currently stands. Therefore, we invite you to submit a revised version of the manuscript that addresses the points raised during the review process.

We look forward to receiving your revised manuscript.

Kind regards,

Luca Navarini

Academic Editor

PLOS ONE

Journal Requirements:

Reviewers' comments:

Reviewer's Responses to Questions

**Comments to the Author**

1. Is the manuscript technically sound, and do the data support the conclusions?

Reviewer #1: Yes

Reviewer #2: Partly

2. Has the statistical analysis been performed appropriately and rigorously? 

Reviewer #1: I Don't Know

Reviewer #2: Yes

3. Have the authors made all data underlying the findings in their manuscript fully available?

Reviewer #1: Yes

Reviewer #2: Yes

4. Is the manuscript presented in an intelligible fashion and written in standard English?

Reviewer #1: Yes

Reviewer #2: Yes

5. Review Comments to the Author

Reviewer #1: I strongly recommend to accept this paper that focuses the tension between public health ethics and clinical ethics. I would also suggest the authors to consider the fact that also rheumatic diseases constitute a public health object, therefore there should not be any conflict between them, whereas it happens it means a poor understanding of the very concept of public health.

Reviewer #2: The authors presented a very well-organized paper on the effects of the SARS-CoV2 pandemic on rheumatic patients. However, some changes are required:

line 108 there is a code (NCTO45573589). What is it for?

Line 241: How the Authors compared disease activity before and after the missed visit? Do they used the same score systems Could the Authors clarify the symptoms and the laboratory markers used to evaluate the “disease activity”? How did these items change for the different diseases? Do they relate to any validated score? Which diseases are listed in “other rheumatologic diseases? Could they clarify if the disease activity assessment relies only on PROs?

Line 319: The Authors should better discuss their results.

Line 324: Long disease duration, substantial disease activity, need for rheumatic medical care, and

NP with rheumatic disease-associated treatment was associated with increased disease activity. Where are these results?

How did face-to-face visits compare to telemedicine affected disease control?

The discussion is too long and sometimes is unfocused

English should be improved

6. PLOS authors have the option to publish the peer review history of their article (what does this mean?). If published, this will include your full peer review and any attached files.

Reviewer #1: **Yes: **Giampaolo Ghilardi

Reviewer #2: No

---

## [Author Response · Author response to Decision Letter 0]

13 Apr 2021

RESPONSES TO EDITOR AND REVIEWERS

 Response

We have revised PLOS ONE´s manuscript style requirements.

Response 

We have revised the references list. We added the DOI number to all the references and complete a published media reference. We did not identify any cited papers that have been retracted.

Reviewer #1: I strongly recommend to accept this paper that focuses the tension between public health ethics and clinical ethics. I would also suggest the authors to consider the fact that also rheumatic diseases constitute a public health object, therefore there should not be any conflict between them, whereas it happens it means a poor understanding of the very concept of public health.

Response

We really appreciate the comment. We agree with the reviewer, chronic diseases such as rheumatic diseases are object of public health. Our specialty cares for people with rheumatic and inflammatory diseases that affect the joints and connective tissues; nonetheless, as clinicians, rheumatologists work at the bed-side, and take care of individuals who are affected by a rheumatic disease; we adopt a patient-centered approach, that considers unique patients and their unique circumstances, priorities and needs; we build patient-doctor relationships with patients, at the individual level, and hardly consider the population level, which is the matter of public health. 

The tension between public health ethic and bed-side clinical ethics certainly needs a more comprehensive approach, but we have been required to shorten and focus the discussion. 

Reviewer #2: The authors presented a very well-organized paper on the effects of the SARS-CoV2 pandemic on rheumatic patients. However, some changes are required:

Response 

Thank you for the comment. 

Line 108 there is a code (NCT04557358). What is it for?

Response 

We apologize for the information missed. We have added that the study was registered at clinicaltrials.gov (largest clinical trials database, which is run by United States Library of Medicine at the National Institutes of health). NCT04557358 is the identifier assigned when the current clinical trial was registered. 

Line 241: How the Authors compared disease activity before and after the missed visit? Do they used the same score systems

Response 

We have added the following paragraph to be more precise, in the- Assessment of the clinical status of rheumatic disease section: “In addition, an independent observer who was a certified rheumatologist, reviewed all the charts from the patients included, and scored the categories related to the clinical status of the underlying rheumatic disease, according to the pre-specified criteria summarized in table 2, twice, at the last consultation before HCI and at the first face-to face consultation after the OCDIR was reinstalled. Pre-specified criteria included symptoms, clinical findings and serological markers, available in the medical charts; these were identified, analyzed and integrate by the independent observer, who additionally considered the specific diagnosis of the underlying rheumatic disease.” 

Table 2 summarizes the pre-specified criteria used to define the subcategories, within each of the 3 categories related to the clinical status of the rheumatic disease.

Could the Authors clarify the symptoms and the laboratory markers used to evaluate the “disease activity”? How did these items change for the different diseases? Do they relate to any validated score? 

Response:

No validated scores were used, as they usually relied on serological markers that were not consistently available on the medical charts. Instead, disease activity status was defined based on the independent observer judgment (a certified rheumatologist) who integrated (available) symptoms, clinical findings and serological marker and considered the underlying rheumatic diagnosis; this approach reflects current daily practice, where the use of validated tools is exceptional and limited to research purposes. 

We propose the paragraph described in the previous query. 

Which diseases are listed in “other rheumatologic diseases? 

Response

We have added a supplementary table (Supplementary table 1. N° (%) of patients with a face-to-face consultation at the OCDIR, with the ten most frequent rheumatic diagnoses specified) that depicts the patients’ distribution of the most frequent underlying rheumatic diagnoses. 

Could they clarify if the disease activity assessment relies only on PROs?

Response

Unfortunately PROs were not included as part of the assessment of the patients disease activity status. PROs regularly require time from the patients and formats to record the PROs, and both were limited when the outpatient clinic was reinstalled for face-to-face consultations. A better description of how disease activity was assessed, is provided in the corresponding section and table 2. 

Line 319: The Authors should better discuss their results.

Response

We propose the following paragraph: “Meanwhile, Ciurea et al. [3] demonstrated that a transient reduction in the rheumatology services during the COVID-19 pandemic had no detrimental impact on the disease course in 666 Swiss patients with axial spondyloarthritis, RA and psoriatic arthritis; however, the authors reported a 129% increase in the number of remote assessments, in an effort to compensate for the drop of the face-to face consultations, that might have influenced patients’ outcomes”.

Line 324: Long disease duration, substantial disease activity, need for rheumatic medical care, and NP with rheumatic disease-associated treatment was associated with increased disease activity. Where are these results?

Response

We apologize for the misunderstanding due to the lack a consistency when describing the sub-categories related to the current course of disease activity. We have been more precise and consistent with the terms used all along the text. In particular “increased disease activity” has been changed to “clinical deterioration” in the manuscript. 

How did face-to-face visits compare to telemedicine affected disease control?

Response

The study was not designed to answer the question; in addition, remote consultations do not consistently assess and record the level of the underlying rheumatic disease activity. 

The discussion is too long and sometimes is unfocused

Response 

We have shortened the discussion (20%) and tried to focus on the relevant clinical and ethical aspects. 

English should be improved

Response

Before submitting the manuscript, it was sent to a professional editing service for English language style review and a certificate was issued by the Wiley editing Service Company. We have additionally revised the updated version.

---

## [Decision Letter · Decision Letter 1]

11 Jun 2021

Clinical and bioethical implications of health care interruption during the COVID-19 pandemic: A cross-sectional study in outpatients with rheumatic diseases

PONE-D-21-03232R1

Dear Dr. Pascual-Ramos,

We’re pleased to inform you that your manuscript has been judged scientifically suitable for publication and will be formally accepted for publication once it meets all outstanding technical requirements.

Kind regards,

Luca Navarini

Academic Editor

PLOS ONE

Additional Editor Comments (optional):

Reviewers' comments:

Reviewer's Responses to Questions

**Comments to the Author**

1. If the authors have adequately addressed your comments raised in a previous round of review and you feel that this manuscript is now acceptable for publication, you may indicate that here to bypass the “Comments to the Author” section, enter your conflict of interest statement in the “Confidential to Editor” section, and submit your "Accept" recommendation.

Reviewer #1: All comments have been addressed

2. Is the manuscript technically sound, and do the data support the conclusions?

Reviewer #1: Yes

3. Has the statistical analysis been performed appropriately and rigorously? 

Reviewer #1: I Don't Know

4. Have the authors made all data underlying the findings in their manuscript fully available?

Reviewer #1: Yes

5. Is the manuscript presented in an intelligible fashion and written in standard English?

Reviewer #1: Yes

6. Review Comments to the Author

Reviewer #1: I've not the necessary knowledge to assess the statistical analysis.

The authors have successfully addressed the questions I asked.

7. PLOS authors have the option to publish the peer review history of their article (what does this mean?). If published, this will include your full peer review and any attached files.

Reviewer #1: **Yes: **Giampaolo Ghilardi

---

## [Editor Report · Acceptance letter]

28 Jun 2021

PONE-D-21-03232R1 

Clinical and bioethical implications of health care interruption during the COVID-19 pandemic: A cross-sectional study in outpatients with rheumatic diseases 

Dear Dr. Pascual-Ramos:

I'm pleased to inform you that your manuscript has been deemed suitable for publication in PLOS ONE. Congratulations! Your manuscript is now with our production department. 

Kind regards, 

on behalf of

Dr. Luca Navarini 

Academic Editor

PLOS ONE